# REVIP: Rethinking Visual Prompting for Multimodal Large Language Models with External Knowledge

## Abstract

In recent years, multimodal large language models (MLLMs) have made significant strides by training on vast high-quality image-text datasets, enabling them to generally understand images well. However, the inherent difficulty in explicitly conveying fine-grained or spatially dense information (e.g., object masks) in the text format poses a challenge for MLLMs, limiting their ability to answer questions requiring an understanding of detailed or localized visual elements. Drawing inspiration from the Retrieval-Augmented Generation (RAG) concept, this paper proposes a new visual prompt approach to integrate fine-grained external knowledge, obtained from specialized vision models (*e.g.*, instance segmentation/OCR models), into MLLMs. This is a promising yet underexplored direction for enhancing MLLMs' performance. Our approach diverges from concurrent works, which transform external knowledge into additional text prompts, necessitating the model to indirectly learn the correspondence between visual content and text coordinates. Instead, we propose embedding fine-grained object knowledge directly into a spatial embedding map as a visual prompt. This design can be easily incorporated into various MLLMs, such as LLaVA and Mipha, considerably improving their visual understanding performance. Through rigorous experiments, we demonstrate that our method can enhance MLLM performance across 11 benchmarks, improving their fine-grained context-aware capabilities.

## 1 Introduction

The advancement of large language models (LLMs) [1, 2, 3, 4] has revolutionized how machines process and generate human-like text, demonstrating remarkable abilities in reasoning, translation, and contextual understanding. The integration of language and vision into unified models, such as GPT-4V [5], represents a significant leap forward in enabling machines to understand and interact with the world in a manner akin to human cognition.

Despite their remarkable capabilities, most of the MLLMs (shown in Figure 1 (a)) trained with image-text pairs still often struggle in fine-grained multimodal comprehension capacities, *e.g.*, correctly counting objects or precisely locating a specific object. This is partially because of the lack of high-quality data with fine-grained text descriptions. Moreover, text itself has inherent limitations in accurately conveying fine-grained spatial information. As a result, current MLLMs often fail to accurately interpret pixel-level visual content of localized regions within an image, which in return impacts the overall comprehension capacity and thereby causing the notorious "hallucination" problem [6].

To tackle this challenge, one line of work [7, 8, 9] explicitly integrates region coordinates information into the text prompt and trains on specialized region-level chat data. However this still demands that the model implicitly learns to understand coordinates and establish connections with visual content, thereby increasing the learning complexity. Another line of work [10, 11, 12] proposes incorporating Region of Interest (ROI) features directly into model learning, necessitating bespoke model architectures. In contrast to these approaches, rather than learning region information from scratch, this paper explores leveraging finely-grained recognition predictions from existing vision models as external knowledge for MLLMs, inspired by the RAG concept. Concurrent with our

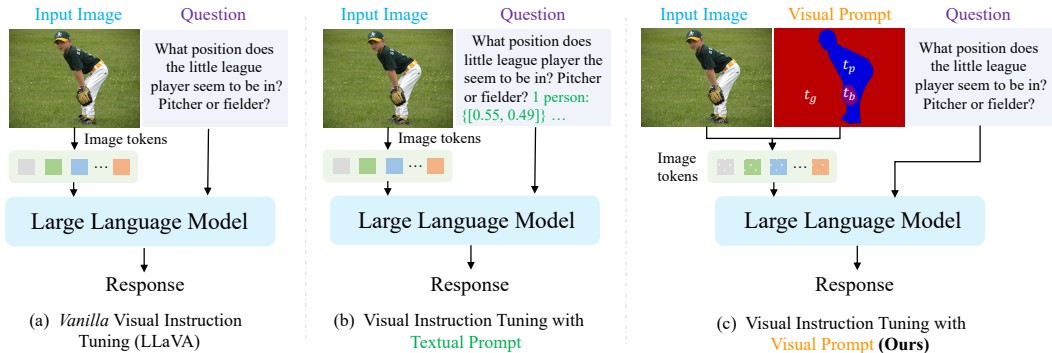

Figure 1: **Different training paradigms.** (a) means the original visual instruction tuning of LLaVA [16]. (b) denotes visual instruction tuning with external textual prompts [13] (*e.g.*, 1 person and the center coordinates of its bounding box: [0.55,0.49]), note that we neglect the template prefix of textual prompts for visualization. (c) is the proposed auxiliary visual prompt, which is a feature map composed with different object regions. For each pixel, it is filled out with the textual embedding of the corresponding *categories* or *OCR text* ($t_g$, $t_p$ and $t_b$ in the example visual prompt mean the textual embeddings of *grass*, *person* and *baseball glove*).

work, one recent approach [13] introduces external knowledge, such as regional coordinates from object detection and Optical Character Recognition (OCR) technologies, into MLLMs (shown in Figure 1 (b)), helping understand localized multimodal content. However, this method still integrates external knowledge through the text prompt, requiring implicit learning of content-to-coordinate correspondence by the model. Furthermore, it lacks support for more nuanced external knowledge, such as instance masks.

In this paper, we propose a new visual prompt paradigm to inject external knowledge, such as localized information, into MLLMs, addressing the challenge of precisely aligning detailed content across multiple modalities. As illustrated in Figure 1 (c), the core idea is, rather than treating local context information as a part of text prompts, we embed them directly within the visual prompts. Specifically, we start by leveraging panoptic segmentation [14] and OCR detection [15] models, and a pre-trained text encoder to generate pixel-wise text embeddings, which are served as the local context information for MLLMs. Subsequently, we extend the original visual prompts by adding the newly generated context information in a spatial-wise manner. This integrated prompt is then assimilated into MLLMs, improving fine-grained visual content comprehension. Consequently, our approach is capable of enabling MLLMs to discern contexts in the pixel-level space and improve their performance.

With the proposed visual prompt paradigm, we train a set of MLLMs on the LLaVA-1.5 datasets [16]. The experimental results show that, even with 3 billion parameters, our method improves upon the leading open-source MLLMs such as LLaVA-1.5 [17, 16] and Qwen-VL [18], without collecting additional chat data for training. Remarkably, our models show superior performance across a wide array of benchmarks when compared to the 7-billion MLLM variants, including LLaVA-1.5, Qwen-VL, and InstructBLIP [19], and in some instances, even outperform their 13-billion MLLM counterparts. Our experimental results confirm the significance of integrating our proposed prompt approach with MLLMs to enhance their capabilities.

The contributions can be summarized as follows:

- We systematically investigate integrating localized information into MLLMs. Empirical findings suggest that our proposed visual prompt significantly outperforms the previous prompt paradigm relying solely on textual prompts containing coordinates.

- We propose to integrate contextual embeddings within local contours (*e.g.*, object masks) as the visual prompt, which facilitates the establishment of correlations between image pixels and contexts, thereby enhancing the fine-grained understanding capabilities of various MLLMs across a spectrum of benchmarks.

- Based on our proposed approach, our model with 3B parameters surpasses or achieves comparable performances with both existing 7B and 13B models across 11 benchmarks.

## 2 RELATED WORK

**Large Language Models.** The initial potential of large language models (LLMs) was showcased by foundational works like BERT [20] and GPT [21]. They sparked a wave of scaling efforts, leading to a range of influential projects, such as T5 [22], GPT-3 [23], Flan-T5 [24], and PaLM [25]. As the volume of training data expanded and the dimensions of model parameters grew, these scaling endeavors led to the creation of ChatGPT [26, 27]. Models like LLaMA [1] and GPT-4 [3] have been trained on extensive corpora and demonstrated remarkable capabilities in diverse cognitive tasks. Additionally, lightweight LLMs with fewer than 3B parameters, *i.e.*, Phi [28, 29] and StableLM-2 [30] have shown performance comparable to larger models [31]. In our work, we adopt Phi-2 [29], Vicuna-7B [31] and Vicuna-13B [31] as our language backbone.

**Multimodal Large Language Models.** Influenced by the success of instruction tuning from LLM, LLaVA [17] and MiniGPT-4 [32] have adopted visual instruction tuning to improve LLMs' interaction with visual data, yielding impressive outcomes. Kosmos-2 [33] and Shikra [34] have advanced MLLMs by enhancing visual comprehension capabilities. Works like LLaVA-Phi [35], MobileVLM [36] and Bunny [37] mainly focus on optimizing training recipes and architecture design for lightweight MLLMs. V* [38] searches visual targets using LLMs' contextual cues to enhance MLLM's performance. To solve the challenge of understanding fine-grained information in images, existing approaches propose to learn coordinate representations [7, 34, 8] and Region of Interest (ROI) features [33, 11], which use inflexible visual referral formats or necessitate the collection of region-level training data. On the contrary, we focus on utilizing external knowledge to improve the fine-grained vision-language alignment for MLLMs without collecting extra chatting data.

**Prompting Multimodal Large Language Models.** Inspired by the ability of GPT-4V [5] to process diverse inputs, ViP-LLaVA [9] collects a visual prompt instruction dataset containing various visual prompts, *e.g.*, scribbles and arrows, for MLLMs fine-tuning. [39] proposes to incorporate the cropped regions to enhance the performance of MLLMs. Contemporary to our work, [13] has offered advanced insights in prompting MLLMs through external knowledge, which introduces bounding box and OCR coordinates into text prompt, however, it's still challenging to interpret the pixel-level contexts. In this paper, we investigate how to efficiently utilize external knowledge to enhance multimodal fine-grained alignment of MLLMs and introduce a novel visual prompt paradigm incorporating pixel-level contextual information.

## 3 PROPOSED METHOD

In this section, we propose a new visual prompt paradigm that integrates local external information to enhance the capability of MLLMs. In section 3.1, we outline the design of the auxiliary visual prompt that contains detailed region-specific information. Using the auxiliary visual prompt, in section 3.2, we further embed it into MLLMs by merging it with the original visual tokens. Finally, we briefly introduce the details of training in section 3.3.

### 3.1 AUXILIARY VISUAL PROMPT WITH EXTERNAL KNOWLEDGE

In this section, we propose a method to generate local contextual external knowledge to assist MLLMs. In contrast to [13], which focuses solely on object detection and OCR information and integrates them as part of the text prompt, we enhance the granularity of local external knowledge by leveraging a panoptic segmentation model, it provides comprehensive pixel-level annotations that include both object instances and background, offering detailed scene understanding. Additionally, we continue to utilize an OCR model but transform both types of external knowledge into pixel-wise embeddings. Further details are provided below.

As shown in Figure 2, given the input image $I \in \mathbb{R}^{3 \times H \times W}$, we can obtain the granular pixel-level information by an off-the-shelf panoptic segmentation model [14] and an OCR model [15]. The generation of the external knowledge can be expressed as:

$$\{M_j, C_j\}_{j=1}^{N_s} = f_{\text{seg}}(I), \quad \{B_j, T_j\}_{j=1}^{N_o} = f_{\text{ocr}}(I), \tag{1}$$

Figure 2: **Auxiliary visual prompt generation**. It firstly generates the panoptic segmentation masks [14] for the input image, there's a class category for each mask region, then we can obtain the textual embeddings (*e.g.*, $t_{\mathrm{book}}$, $t_{\mathrm{bed}}$ and $t_{\mathrm{cat}}$) through a pre-trained text encoder for all the classes (*e.g.*, book, bed, cat). Finally, the auxiliary visual prompt can be generated by concatenating these textual embeddings within the corresponding mask regions together. Note that we can also adopt the OCR model [15] to obtain the texts and the regions, we don't display it here for clearer explanation.

where $f_{\mathrm{seg}}(\cdot)$ and $f_{\mathrm{ocr}}(\cdot)$ mean panoptic segmentation and optical character recognition (OCR) models, $N_{\mathrm{s}}$ and $N_{\mathrm{o}}$ are the numbers of detected mask regions and OCR bounding boxes. $\{M_j, C_j\}_{j=1}^{N_{\mathrm{s}}}$ is the set of mask regions and the corresponding classes, and $\{B_j, T_j\}_{j=1}^{N_{\mathrm{o}}}$ represents the set of detected OCR bounding boxes and texts.

With the detected classes $\{C_j\}_{j=1}^{N_{\mathrm{s}}}$ and OCR texts $\{T_j\}_{j=1}^{N_{\mathrm{o}}}$, a pre-trained text encoder ($f_{\mathrm{text}}(\cdot)$) is leveraged to generate the textual embeddings as:

$$
\begin{aligned}
\mathcal{T}_{\mathrm{s}} &= \{t_1, \ldots, t_{N_{\mathrm{s}}}\} = \{f_{\mathrm{text}}(C_1), \ldots, f_{\mathrm{text}}(C_{N_{\mathrm{s}}})\}, \\
\mathcal{T}_{\mathrm{o}} &= \{\hat{t}_1, \ldots, \hat{t}_{N_{\mathrm{o}}}\} = \{(f_{\mathrm{text}}(T_1), \ldots, f_{\mathrm{text}}(T_{N_{\mathrm{o}}})\},
\end{aligned}
\tag{2}
$$

where $t_i \in \mathbb{R}^{1 \times d}(1 \leqslant i \leqslant N_{\mathrm{s}})$ and $\hat{t}_i \in \mathbb{R}^{1 \times d}(1 \leqslant i \leqslant N_{\mathrm{o}})$ denote the $i_{\mathrm{th}}$ textual embedding vector of the classes for the detected mask region and OCR texts respectively, while $d$ is the embedding dimension.

In order to generate a pixel-wise visual prompt for the external knowledge instead of a pure text description for the regions with coordinates and category names, the auxiliary visual prompt is initialized as a zero tensor $\mathcal{P} \in \mathbb{R}^{H \times W \times d}$ and then filled with the newly generated textual embeddings for the external knowledge as:

$$
\begin{aligned}
\mathcal{P}_{j,k} &= \begin{cases} t_u & \text{if } (j,k) \in M_u \\ \mathcal{P}_{j,k} & \text{otherwise} \end{cases} \quad \forall u \in \{1, \ldots, N_{\mathrm{s}}\}, \\
\mathcal{P}_{j,k} &= \mathcal{P}_{j,k} + \begin{cases} \hat{t}_v & \text{if } (j,k) \in B_v \\ 0 & \text{otherwise} \end{cases} \quad \forall v \in \{1, \ldots, N_{\mathrm{o}}\}.
\end{aligned}
\tag{3}
$$

Note, for some regions, if the confidence of the class prediction given by the segmentation model is low or the OCR model fails to detect any text, we leave the region area with zero values. For the regions that are occupied by both models, we simply add the text embeddings directly. We leave the investigation of more refined fusion techniques to future research.

With the auxiliary visual prompt containing pixel-level local contextual information from panoptic segmentation and OCR models, MLLMs can effectively capture finer-grained features. The next challenge is to establish a clearer connection between these pixel-wise annotations and the original image feature. This will help alleviate the model's difficulties in learning their relationship effectively.

## 3.2 VISUAL PROMPT INFUSION

In this section, we introduce the visual prompt infusion that incorporates the proposed auxiliary visual prompts into the MLLMs. Previous methods [13] choose to append the external knowledge (embeddings for object category and its coordinates) to the text prompts, which requires the model to

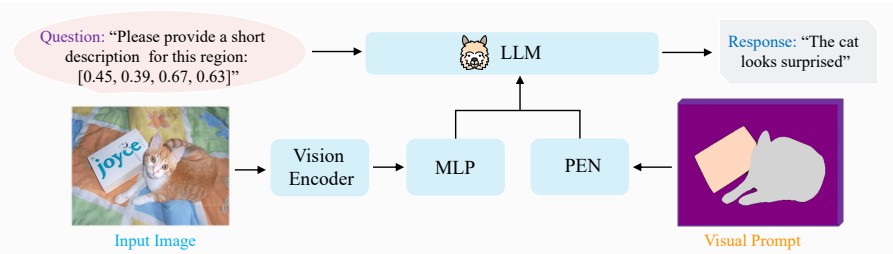

Figure 3: **The illustration of visual instruction tuning with the generated visual prompt.** Our proposed visual prompt can be easily combined with existing multimodal large language models (*e.g.*, LLaVA [16]), note that *PEN* means prompt embedding network.

learn the correspondence of visual content within the specified coordinates encoded in the external knowledge and, as a result, increasing the difficulties of the learning process of the model. To address this challenge, we propose to directly align the auxiliary visual prompt with the image features on a pixel-by-pixel basis.

Specifically, as shown in Figure 3, the image tokens are first generated via an image encoder $f_{\text{img}}(\cdot)$ and an MLP projector ($f_{\text{MLP}}(\cdot)$):

$$\mathcal{F}_{\text{v}} = f_{\text{MLP}}(f_{\text{img}}(I)), \tag{4}$$

where $\mathcal{F}_{\text{v}} \in \mathbb{R}^{N_{\text{v}} \times d_{\text{v}}}$, $N_{\text{v}}$ and $d_{\text{v}}$ represent the number of image tokens and the embedding dimension. Then, the auxiliary visual prompt is further processed by a prompt embedding network (PEN) as

$$\mathcal{F}_{\text{p}} = f_{\text{PEN}}(\mathcal{P}). \tag{5}$$

For the prompt embedding network, we employ three convolutional layers, with an activation layer (ReLU) inserted between each pair of them. This network primarily serves to align the feature space and spatial size between the image tokens and the auxiliary visual prompts.

When combining the image tokens and the processed auxiliary visual prompt, we mainly consider two options, both of which operate pixel-wise. **(1) feature fusion:** $\hat{\mathcal{F}}_{\text{v}} = f(\text{Concat}(\mathcal{F}_{\text{v}}, \mathcal{F}_{\text{p}}))$, where $f$ is a linear layer that maps the embedding $\mathbb{R}^{N_v \times d_{2v}} \rightarrow \mathbb{R}^{N_v \times d_v}$ to maintain the total number of image tokens unchanged; **(2) feature addition**, $\hat{\mathcal{F}}_{\text{v}} = \mathcal{F}_{\text{v}} + \mathcal{F}_{\text{p}}$, which sums the two types of features directly.

The advantages of the pixel-wise fusion for both options facilitate correspondence between external knowledge and original visual features. Providing explicit pixel labels for segmentation and OCR allows the model to easily interpret pixel categories and associated OCR text descriptions. This guidance is crucial in helping the model disambiguate complex scenes, highlight salient features, and distinguish finer objects, thereby improving its overall performance.

### 3.3 TRAINING

Training MLLMs involves predicting responses based on multimodal inputs using an autoregressive approach. The objective is to maximize the probability of generating tokens that match the ground-truth answer $Y_{\text{a}}$. With the new visual embedding $\hat{\mathcal{F}}_{\text{v}}$, this can be mathematically expressed as follows:

$$P(Y_{\text{a}}|\hat{\mathcal{F}}_{\text{v}}, \mathcal{F}_{\text{t}}) = \prod_{i=1}^{L} P_{\boldsymbol{\theta}}(y_i|\hat{\mathcal{F}}_{\text{v}}, \mathcal{F}_{\text{t}}, Y_{\text{a},<i}). \tag{6}$$

Here, $L$ represents the sequence length of the ground truth answer $Y_{\text{a}}$, $\boldsymbol{\theta}$ means the trainable parameters. $Y_{\text{a},<i}$ represents all the answer tokens preceding the current prediction token $x_i$, where $i$ denotes the step in the sequence of text token generation. $\mathcal{F}_{\text{t}} \in \mathbb{R}^{N_{\text{t}} \times d_{\text{t}}}$ is the token embedding of the input question, $N_{\text{t}}$ and $d_{\text{t}}$ denote the number of text tokens and token embedding dimension. By fusing these enriched visual cues with the training pipeline, MLLMs can develop a more comprehensive

Table 1: The ablation study of different prompting methods. *Mipha-3B* is the baseline with standard visual & text prompt. *Mipha-3B+FTBI* denotes using textual prompting with Fine-tuning Based Infusion (FTBI) [13]. REVIP-FF and REVIP-FA mean *feature fusion* and *feature addition* respectively, which represent two visual prompt fusion methods we use to insert the auxiliary visual prompt to the original image features.

| Method | VQAv2 | GQA | VisWiz | SQA$^I$ | VQA$^T$ | MME-P | MME-C | MMB | MM-Vet | POPE | MMMU |
|---|---|---|---|---|---|---|---|---|---|---|---|
| Mipha-3B | 81.3 | 63.9 | 45.7 | 70.9 | 56.6 | 1488.9 | 295.0 | 69.7 | 32.1 | 86.7 | 32.5 |
| w/ FTBI | 81.6↑ | 62.6↓ | 45.8↑ | 71.4↑ | 57.8↑ | 1472.3↓ | 356.8↑ | 71.0↑ | 34.8↑ | 88.5↑ | 32.8↑ |
| w/ REVIP-FF | 81.9↑ | 64.8↑ | 46.6↑ | 71.6↑ | 57.6↑ | 1493.5↑ | 345.5↑ | 71.3↑ | 34.3↑ | 88.5↑ | 33.2↑ |
| **w/ REVIP-FA** | **82.4**↑ | **65.3**↑ | **47.0**↑ | **71.8**↑ | **57.8**↑ | **1501.2**↑ | **369.1**↑ | **71.5**↑ | **35.1**↑ | **88.7**↑ | **33.5**↑ |

Table 2: The ablation study of using different vision encoders, *i.e.*, SigLIP [42] *v.s.* CLIP [44]. Note that the reported results for *Mipha-3B* using the CLIP vision encoder are from [40].

| Method | Vis Enc | VQAv2 | GQA | VisWiz | SQA$^I$ | VQA$^T$ | MME-P | MME-C | MMB | MM-Vet | POPE | MMMU |
|---|---|---|---|---|---|---|---|---|---|---|---|---|
| Mipha-3B | CLIP | 78.6 | 62.3 | - | 68.2 | 53.0 | - | - | 68.4 | 31.0 | 86.9 | - |
| **Mipha-3B$^+$(Ours)** | **CLIP** | 79.7↑ | 63.7↑ | 45.8 | 70.1↑ | 54.8↑ | 1445.5 | 308.4 | 70.1↑ | 33.7↑ | 88.8↑ | 32.3 |
| Mipha-3B | SigLIP | 81.3 | 63.9 | 45.7 | 70.9 | 56.6 | 1488.9 | 295.0 | 69.7 | 32.1 | 86.7 | 32.5 |
| **Mipha-3B$^+$(Ours)** | **SigLIP** | 82.4↑ | 65.3↑ | 47.0↑ | 71.8↑ | 57.8↑ | 1501.2↑ | 369.1↑ | 71.5↑ | 35.1↑ | 88.7↑ | 33.5↑ |

understanding of visual content, leading to better alignment between visual and textual representations. To accelerate the training process, we follow FTBI [13] to perform fine-tuning on Mipha-3B [40] and LLaVA-1.5 [16] using LoRA [41].

## 4 EXPERIMENT

In this section, we conduct a comprehensive comparison of our method with existing state-of-the-art (SOTA) multimodal models. Additionally, we perform a series of ablation studies to further validate the proposed method. Finally, we provide visualization examples for in-depth analysis.

**Models.** For the vision encoder, we adopt SigLIP-384px [42] for experiments. We leverage Phi-2-2.7B [29], Vicuna-7B [31] and Vicuna-13B [31] model as the language decoder. For the multimodal projector, same as LLaVA [16], we adopt a two-layer MLP. We use OpenSeed [14] and PaddleOCRv2 [15] to generate the per-pixel externally knowledge for pixel class and OCR text, and leverage UAE-Large-V1 [43] to extract the textual embedding.

**Training Setting.** We fine-tune the models on LLaVA-Instruct-150K dataset [16] using LoRA [41] for 1 epoch, at a learning rate of 2e-4 and a batch size of 256 on 32×V100 32GB GPUs. For the setting of LoRA, we set LoRA rank to be 128 and LoRA's hyperparameter $\alpha$ as 256. Note that we fix all the weights of pre-trained modules, *i.e.*, vision encoder, language encoder and MLP, during training. Our models' weights are initialized from Mipha-3B [40], LLava-7B [16] and LLava-13B [16].

**Benchmarks and Baselines.** We evaluate our approach using 11 popular benchmarks to comprehensively assess its multimodal capabilities. These benchmarks include: VQA-v2 test-dev split [45], VisWiz [46], GQA test-dev-balanced split [47], ScienceQA-IMG test split [48], MME perception [49], MME cognition [49], MMBench test split [50], MM-Vet test split [51], TextVQA [52], POPE [53] and MMMU test split [54].

We compare our results with a bunch of state-of-the-art multimodal large language models (MLLMs): BLIP-2 [55], InstructBLIP [19], Shikra-13B [34], IDEFICS80/9B [56], Qwen-VL [18], mPLUG-Owl2 [57], LLaVA-v1.5-13/7B [16], FTBI-13B/7B [13], and multimodal small language models (MSLMs) [40]: MobileVLM [36], LLaVA-Phi [35], MC-LLaVA [58], Imp-v1 [59], MoE-LLaVA-3.6B [60], TinyLLaVA-share-Sig-Phi [61], Bunny [37] and Mipha [40].

### 4.1 ABLATION STUDIES

In this section, we conduct an ablation study to assess the effectiveness of the proposed approach. By default, the experiments are conducted using Mipha-3B [40] with Phi-2 [29] as the language backbone unless otherwise specified. Note that we use Mipha-3B$^+$ to denote Mipha-3B using our presented REVIP method.

Table 3: The ablation study of adopting different textual encoders, *i.e.*, CLIP [44] *v.s.* UAE [43], to extract textual embeddings for the proposed visual prompt.

| Method | Text Enc | VQAv2 | GQA | VisWiz | SQA$^I$ | VQA$^T$ | MME-P | MME-C | MMB | MM-Vet | POPE | MMMU |
|---|---|---|---|---|---|---|---|---|---|---|---|---|
| Mipha-3B | - | 81.3 | 63.9 | 45.7 | 70.9 | 56.6 | 1488.9 | 295.0 | 69.7 | 32.1 | 86.7 | 32.5 |
| Mipha-3B$^+$(Ours) | CLIP | 82.1↑ | 64.9↑ | 46.2↑ | 71.3↑ | 57.4↑ | 1497.2↑ | 361.5↑ | 71.1↑ | 34.6↑ | 88.5↑ | 33.1↑ |
| **Mipha-3B$^+$(Ours)** | **UAE** | **82.4↑** | **65.3↑** | **47.0↑** | **71.8↑** | **57.8↑** | **1501.2↑** | **369.1↑** | **71.5↑** | **35.1↑** | **88.7↑** | **33.5↑** |

Table 4: The ablation study of utilizing different external knowledge, "Seg" and "OCR" denote panoptic segmentation and OCR information respectively.

| Seg | OCR | VQAv2 | GQA | VisWiz | SQA$^I$ | VQA$^T$ | MME-P | MME-C | MMB | MM-Vet | POPE | MMMU |
|---|---|---|---|---|---|---|---|---|---|---|---|---|
| ✗ | ✗ | 81.3 | 63.9 | 45.7 | 70.9 | 56.6 | 1488.9 | 295.0 | 69.7 | 32.1 | 86.7 | 32.5 |
| ✓ | ✗ | 81.9↑ | 64.7↑ | 46.5↑ | 71.3↑ | 57.1↑ | 1498.3↑ | 355.2↑ | 70.8↑ | 34.0↑ | 87.9↑ | 33.0↑ |
| ✓ | ✓ | **82.4↑** | **65.3↑** | **47.0↑** | **71.8↑** | **57.8↑** | **1501.2↑** | **369.1↑** | **71.5↑** | **35.1↑** | **88.7↑** | **33.5↑** |

Table 5: The comparison of adopting the external knowledge via different visual prompts.

| Method | VQAv2 | MMB | POPE | MM-Vet | SQA$^I$ | MME-P | MME-C | VisWiz | GQA | VQA$^T$ | MMMU |
|---|---|---|---|---|---|---|---|---|---|---|---|
| LLAVA-1.5-7B | 78.5 | 64.3 | 85.9 | 30.5 | 66.8 | 1510.7 | 316.1 | 50.0 | 62.0 | 58.2 | 32.0 |
| w/ clip-CROP | 78.5 | 64.9 | 86.6 | 31.5 | 67.8 | 1465.4 | 345.6 | 50.2 | 62.3 | 58.7 | 32.1 |
| w/ yolo-CROP | 78.4 | 65.1 | 86.8 | 31.4 | 67.6 | 1455.9 | 347.9 | 50.3 | 62.4 | 58.6 | 32.1 |
| w/ sam-CROP | 78.7 | 65.4 | 86.9 | 32.6 | 68.0 | 1478.3 | 352.2 | 50.3 | 62.5 | 58.8 | 32.2 |
| **w/ REVIP (Ours)** | **79.8** | **67.6** | **88.9** | **34.9** | **69.5** | **1515.3** | **399.5** | **51.5** | **63.3** | **59.8** | **33.1** |

**Prompting MLLMs with Different Approaches.** In Table 1, we present the results of the ablation study for four different prompting strategies: (1) Mihpa-3B baselines with vanilla text prompt, as used by LLaVA-1.5 [16]. (2) Mihpa-3B + FTBI proposed in [13] that appends external local contextual knowledge to the text prompts. (3) The proposed auxiliary visual prompt inserted via feature fusion. (4) The proposed auxiliary visual prompt added via feature addition.

From Table 1, we note that compared to the baseline (1) with vanilla prompts, both proposed fusion strategies (3) and (4) exhibit a significant improvement. This suggests that external knowledge is indeed beneficial in enhancing the capabilities of MLLMs. In comparison to Mihpa-3B+FTBI (2), which inserts external local contextual knowledge into the text prompt, (4) outperforms it in 10 out of 11 benchmarks, notably for GQA [47] and MME-P [49]. This implies that explicitly linking external local knowledge to the original visual features reduces the model's learning burden in establishing spatial relationships, consequently enhancing performance. Furthermore, we empirically observe that directly adding auxiliary visual prompts yields slightly better results than concatenation. Therefore, we adopt feature addition as our default setting for subsequent experiments.

**The Effect of Using Different Vision Encoders.** In Table 2, we further ablate the effectiveness brought by different vision encoders, *i.e.*, CLIP [44] *v.s.* SigLIP [42], since the Mipha-3B model with the CLIP vision encoder has not been released, we cite the results from its paper [40]. From the results, we can draw two conclusions. First, for both vision encoders, our methods have consistent improvement compared to the baselines, which validates the stability of our methods. Second, SigLIP emerges as the stronger vision encoder when compared to CLIP. Therefore, we opt to utilize SigLIP as the default vision encoder in subsequent sections.

**The Impact of Adopting Different Textual Encoders.** In Table 3, we perform an ablation study using different textual encoders, *i.e.*, CLIP [44] vs. UAE [43], to extract textual embeddings for the proposed visual prompt. We draw two conclusions from Table 3: (1) Using different textual encoders, the proposed approach consistently outperforms the baseline, demonstrating the robustness of our method. (2) Adopting UAE as the pre-trained textual encoder achieves significantly better performance. Therefore, we choose UAE as the default pre-trained textual encoder in our experiments.

**Exploring the Influence of External Knowledge Components.** In Table 4, we perform the ablation of the utilized external knowledge components, we can conclude that adopting the information from OCR can further improve the model's overall performance, especially, when incorporating OCR information for some text-specific tasks (*e.g.*, TextVQA [52] and MM-Vet [51]), it can achieve remarkable performance boosts.

Table 6: The challenging examples on LLaVA-1.5, our approach can provide reasonable reponses for the tricky questions which need to fully understand the contexts of the images.

| **Visual input example, context-related questions:** | |
|---|---|
| 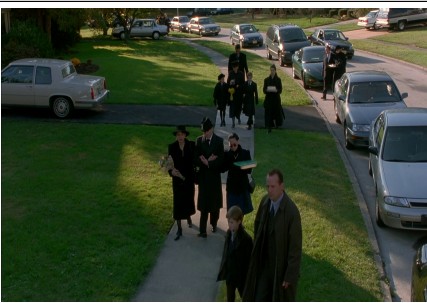 | 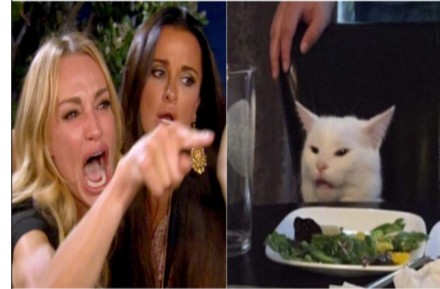 |
| What is funny about this image? | Why are the persons wearing black clothes? |
| **LLaVA-7B** The funny aspect of this image is the juxtaposition of two women pointing fingers at a cat, which is sitting at a table with a plate of food. The cat appears to be looking at the women with an angry expression, as if it is reacting to their pointing fingers. This creates a humorous and unexpected scene, as it is not common to see a cat reacting to human gestures in such a manner. The image captures a light-hearted and amusing moment between the women and the cat. | The persons in the image are wearing black clothes, which is a common attire for religious ceremonies or events. The group of people, including a man and a woman, are walking down the street, possibly participating in a religious procession or a funeral. The black clothing signifies their respect and adherence to the religious or cultural traditions associated with the event they are attending. |
| **Ours** The funny aspect of this image is the contrast between the two women pointing at the cat and the cat's reaction to the situation. One woman is making a funny face, while the cat is sitting on a table with a plate of food, seemingly unbothered by the attention. The cat's calm demeanor in the face of the women's playful behavior creates a humorous scene, as the women try to entertain themselves and the cat seems to be enjoying the attention without reacting to their attempts. | The persons in the image are wearing black clothes, which are typically associated with mourning or a period of grieving. This suggests that they might be attending a funeral or a memorial service for someone who has passed away. The black attire signifies respect and solidarity with the bereaved family and friends during this difficult time. It is a common practice in many cultures to wear black clothes to express sympathy and support for those persons who have experienced a loss. |

**Comparison with Other Visual Prompting Paradigms.** [39] proposes to crop region-of-interests according to the given prompt and utilize the cropped information to further enhance the performances of large language models, we perform the comparison experiments based on LLAVA-1.5-7B [16], which has been presented in Table 5. Specifically, three cropping techniques (clip-CROP, yolo-CROP, and sam-CROP) are employed, following the released code[1]. Note that for fair comparison, we also fine-tune LLAVA-1.5-7B by using the cropped regions with clip/yolo/sam-CROP and report their results. It shows that our method consistently outperforms [39] across all multimodal benchmarks, which can demonstrate the effectiveness of our presented visual prompting method.

## 4.2 MAIN RESULTS

In Table 7, we compare our methods with other state-of-the-art (SOTA) models. We divide the table into sections for language models smaller than 3B and those beyond 7B to provide a clearer comparison. From the results, we observe that our model achieves the best performance on 9 out of 11 benchmarks for larger language models (>7B) and attains the highest accuracy on 9 out of 11 benchmarks for relatively smaller language models (<3B). Note that, in Table 7, some models, *e.g.*, Shikra-13B [34], Qwen-VL [18], are trained with million or billion level data, while our model is only trained on LLaVA-Instruct-150K dataset without collecting any additional chatting data for neither pre-training nor fine-tuning, which highlights the exceptional multimodal understanding and reasoning capabilities of our models. In addition, on top of the LLaVA-1.5 framework, our approach can bring more remarkable and consistent improvement on all benchmarks compared with FTBI [13]. It justifies the proposed infusion strategy, which involves inserting external knowledge in

---

[1]https://github.com/saccharomycetes/visual_crop_zsvqa

Table 7: The comprehensive multi-modal evaluation across 11 distinct benchmarks to thoroughly assess model performance: VQAv2 [45], GQA [47], VisWiz [46], SQA$^I$: ScienceQA-IMG [48], VQA$^T$: TextVQA [52], MME-P: MME Perception [49], MME-C: MME Cognition [49], MMB: MMBench [50], MM-Vet [51], POPE [53] and MMMU [62]. "V", "Q", "L", "M" and "P" mean Vicuna [31], Qwen [18], LLaMA [1], MobileLLaMA [63] and Phi-2 [29]. The image resolution used by the visual backbone is indicated in the column labeled *Res.*, LLaVA-1.5$^+$ and Mipha-3B$^+$ mean LLaVA-1.5 and Mipha-3B models with our presented REVIP method.

| Method | LM | Res. | VQAv2 | GQA | VisWiz | SQA$^I$ | VQA$^T$ | MME-P | MME-C | MMB | MM-Vet | POPE | MMMU |
|---|---|---|---|---|---|---|---|---|---|---|---|---|---|
| Multimodal Large Language Models | | | | | | | | | | | | | |
| BLIP-2 [55] | V-13B | 224 | 65.0 | 41.0 | 19.6 | 61.0 | 42.5 | 1293.8 | 290.0 | - | 22.4 | 85.3 | - |
| InstructBLIP [19] | V-7B | 224 | - | 49.2 | 34.5 | 60.5 | 50.1 | - | - | 36 | 26.2 | - | - |
| InstructBLIP [19] | V-13B | 224 | - | 49.5 | 33.4 | 63.1 | 50.7 | 1212.8 | 291.8 | - | 25.6 | 78.9 | - |
| Shikra [34] | V-13B | 224 | 77.4 | - | - | - | - | - | - | 58.8 | - | - | - |
| IDEFICS-9B [56] | L-7B | 224 | 50.9 | 38.4 | 35.5 | - | 25.9 | - | - | 48.2 | - | - | - |
| IDEFICS-80B [56] | L-65B | 224 | 60.0 | 45.2 | 36.0 | - | 30.9 | - | - | 54.5 | - | - | - |
| Qwen-VL [18] | Q-7B | 448 | 78.8 | 59.3 | 35.2 | 67.1 | 63.8 | - | - | 38.2 | - | - | - |
| Qwen-VL-Chat [18] | Q-7B | 448 | 78.2 | 57.5 | 38.9 | 68.2 | 61.5 | 1487.5 | 360.7 | 60.6 | - | - | 32.9 |
| mPLUG-Owl2 [57] | L-7B | 448 | 79.4 | 56.1 | 54.5 | 68.7 | 58.2 | 1450.2 | 313.2 | 64.5 | 36.2 | 85.8 | 32.1 |
| LLaVA-1.5 [16] | V-7B | 336 | 78.5 | 62.0 | 50.0 | 66.8 | 58.2 | 1510.7 | 316.1 | 64.3 | 30.5 | 85.9 | 32.0 |
| FTBI-7B [13] | V-7B | 336 | 79.0 | 60.5 | - | - | 60.1 | 1482.7 | 397.9 | 67.3 | 35.2 | 88.9 | - |
| **LLaVA-1.5$^+$(Ours)** | **V-7B** | 336 | **79.8↑** | **63.3↑** | **51.5↑** | **69.5↑** | **59.8↑** | **1515.3↑** | **399.5↑** | **67.6↑** | **34.9↑** | **88.9↑** | **33.1↑** |
| LLaVA-1.5 [16] | V-13B | 336 | 80.0 | 63.3 | 53.6 | 71.6 | 61.3 | 1531.3 | 295.4 | 67.7 | 36.1 | 85.9 | 33.6 |
| FTBI-13B [13] | V-13B | 336 | 80.3 | 62.2 | - | - | 61.8 | 1555.1 | 365.4 | 71.4 | 38.9 | 88.8 | - |
| **LLaVA-1.5$^+$(Ours)** | **V-13B** | 336 | **81.3↑** | **64.9↑** | **55.3↑** | **73.5↑** | **63.3↑** | **1568.7↑** | **370.5↑** | **71.3↑** | **39.5↑** | **88.8↑** | **34.8↑** |
| Multimodal Small Language Models | | | | | | | | | | | | | |
| MobileVLM-1.7B [63] | M-1.4B | 336 | - | 56.1 | - | 57.3 | 41.5 | 1196.2 | - | 53.2 | - | 84.5 | - |
| MobileVLM-3B [63] | M-2.7B | 336 | - | 59.0 | - | 61.2 | 47.5 | 1288.9 | - | 59.6 | - | 84.9 | - |
| MobileVLM-v2-1.7B [36] | M-1.4B | 336 | - | 59.3 | - | 66.7 | 52.1 | 1302.8 | - | 57.7 | - | 84.3 | - |
| MobileVLM-v2-3B [36] | M-2.7B | 336 | - | 61.1 | - | 70.0 | 57.5 | 1440.5 | - | 63.2 | - | 84.7 | - |
| LLaVA-Phi [35] | P-2.7B | 336 | 71.4 | - | 35.9 | 68.4 | 48.6 | 1335.1 | - | 59.8 | 28.9 | 85.0 | - |
| MC-LLaVA [58] | P-2.7B | 384 | 64.2 | 49.6 | - | - | 38.6 | - | - | - | - | 80.6 | - |
| Imp-v1 [59] | P-2.7B | 384 | 79.5 | 58.6 | - | 70.0 | 59.4 | 1434.0 | - | 66.5 | 33.1 | 88.0 | - |
| MoE-LLaVA-3.6B [60] | P-2.7B | 384 | 79.9 | 62.6 | 43.7 | 70.3 | 57.0 | 1431.3 | - | 68.0 | 35.9 | 85.7 | - |
| TinyLLaVA [61] | P-2.7B | 384 | 79.9 | 62.0 | - | 69.1 | 59.1 | 1464.9 | - | 66.9 | 32.0 | 86.4 | - |
| Bunny-3B [37] | P-2.7B | 384 | 79.8 | 62.5 | - | 70.9 | - | 1488.8 | 289.3 | 68.6 | - | 86.8 | 33.0 |
| Mipha-3B [40] | P-2.7B | 384 | 81.3 | 63.9 | 45.7 | 70.9 | 56.6 | 1488.9 | 295.0 | 69.7 | 32.1 | 86.7 | 32.5 |
| **Mipha-3B$^+$(Ours)** | **P-2.7B** | 384 | **82.4↑** | **65.3↑** | **47.0↑** | **71.8↑** | **57.8↑** | **1501.2↑** | **369.1↑** | **71.5↑** | **35.1↑** | **88.7↑** | **33.5↑** |

a pixel-wise manner directly into the visual features, as being more effective than appending it to the text prompt [13].

## 4.3 QUALITATIVE RESULT ANALYSIS

We present visualization results in Table 6 and 8 to further illustrate the improvement of our model in terms of both global image understanding and local object and text recognition. Table 6 demonstrates that compared to LLaVA-1.5 7B [16], our approach generates more detailed and contextually relevant responses, *e.g.*, "The cat's calm demeanor in the face of the women's playful behavior" for the left example; "mourning or a period of grieving" and "express sympathy and support for those persons who have experienced a loss" for the right example, which all need a deeper understanding of the global image context. Meanwhile, Table 8 highlights our model's ability to correctly recognize objects' spatial relationships, such as between a "desk lamp" and a "laptop" from the left image, and exhibit stronger OCR capability in detecting words written on a book from the right image, compared to LLaVA-1.5 7B [16]. These visualizations validate the effectiveness of our proposed methods and support the conclusion that incorporating external local contextual information in a spatial-wise manner improves the model's fine-grained recognition capability and enhances its overall ability for global image understanding. Note that we've shown more ablation study experiments and visualization result analysis in the **Appendix**.

**Time Cost and Scalability.** In Figure 4, using the LLaVA-Instruct-150K dataset [16], we report the computational costs, including training and inference times, for both Mipha-3B and Mipha-3B$^+$ (with our proposed REVIP) across four different image resolutions (*i.e.*, 384×384, 512×512, 640×640, and 768×768). Specifically, we resized the input images to these four resolutions. It's worth mentioning that REVIP increases the number of parameters of Mipha-3B only from **3.22B** to **3.23B**. For the input image resolution of 768×768, the training time increases only from **526.4** to **536.2** GPU hours; the inference time per sample increases only from **0.98** to **1.17** seconds, integrating panoptic segmentation and OCR information adds **0.14** and **0.05** seconds, respectively, contributing

Table 8: The challenging examples on LLaVA-1.5. Our approach can generate accurate responses for text-related questions.

| **Visual input example, relationship-aware and text-related questions:** | |
|---|---|
| 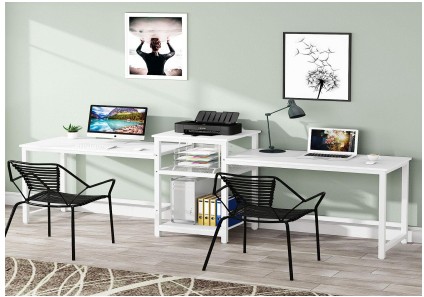 | 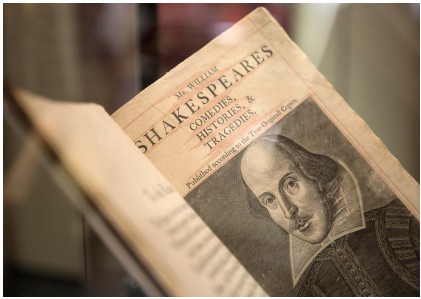 |
| On the right desk, what is to the left of the laptop? | What are all the scene text in the image? |
| **LLaVA-1.5 7B** To the left of the laptop, there is a stack of books on the desk. | The scene text in the image is "Shakespeare's Dramas, Sonnets, & Poems". |
| **Ours** There is a desk lamp to the left of the laptop on the right desk. | The scene text in the image includes the title "Shakespeare's Comedies, Histories, and Tragedies". |

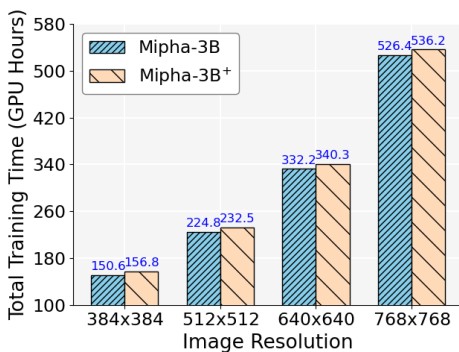
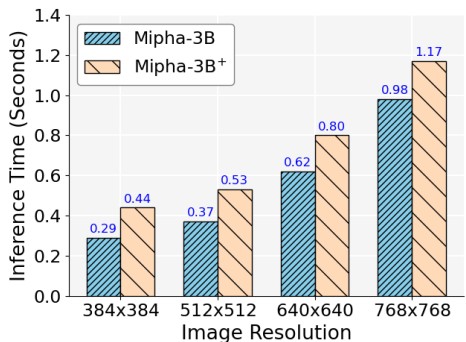

Figure 4: The time costs of Mipha-3B and Mipha-3B$^+$ across a range of image resolutions.

to the total increase of **0.19** seconds. This demonstrates that our method's enhancements come with only a modest computational cost and are even scalable to a $768 \times 768$ image resolution.

## 5 LIMITATIONS AND BROADER IMPACT

Our method relies on pre-trained panoptic segmentation and OCR detection models in a zero-shot fashion, making their performance critical to our approach—especially when substantial domain gaps exist between the benchmark images and their training data.

While our method promises to significantly enhance the cognitive capabilities of multimodal models and inspire new methodologies, users should be aware of potential societal impacts, such as biases arising from training data in MLLMs, segmentation, or OCR models, which may lead to biased responses. However, typical textual prompting methods [13] that incorporate captions, object names, and OCR information for MLLMs can also contain biases or errors.

## 6 CONCLUSION

We propose a method to enhance multimodal language models (MLLMs) by leveraging external knowledge such as localized contextual information. By extracting pixel-wise contextual information using panoptic segmentation and OCR models and integrating it with visual features, our model better understands fine-grained objects and global image context. Experimental results and comparisons with state-of-the-art methods demonstrate our approach's effectiveness. We hope this work highlights the importance of external knowledge for MLLMs and offers an effective way to leverage it.

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
