## A APPENDIX

In the supplementary materials, we provide the following sections:

(a) More implementation details in Section B.
(b) Ablation study experiments in Section C.
(c) Visualization result analysis in Section D.

## B IMPLEMENTATION DETAILS

For the initialization of the proposed prompt embedding network (PEN), we use Kaiming initialization technology [1]. The UAE-Large-V1[1] model is adopted as the pre-trained textual encoder to extract textual embeddings for the visual prompt.

## C ABLATION STUDY

Next, we conduct more ablation study experiments to provide deeper insight into the components of our proposed approach.

**Object Detector v.s. Segmentation Model.** To determine the effect of using an object detector or segmentation model to incorporate pixel-level semantics into the proposed visual prompt, we conduct an ablation study with the popular object detector GroundingDINO [2] and the segmentation model OpenSeed [3]. The results are shown in Table 1. We observe that both GroundingDINO and OpenSeed significantly boost performance across all benchmarks. However, utilizing OpenSeed achieves better performance gains due to its fine-grained mask regions. Thus, we adopt OpenSeed by default to generate object regions.

**The Effect of Fine-Tuning with the Visual Prompt.** As displayed in Table 2, the model fine-tuned with the proposed visual prompt (*i.e.*, the third row) achieves remarkably better performance than the one fine-tuned without our visual prompt (*i.e.*, the second row) across all benchmarks. Specifically, without using our visual prompt for fine-tuning, the model even shows performance degradation on Text-VQA benchmark [4] and has negligible gains on Science-QA [5], VQAv2 [6], MME-P [7], and MME-C [7] benchmarks. All these results demonstrate the superiority of the proposed method.

**Discussion.** We also compare our presented REVIP method with $V^*$ [8], which employs an LLM-guided visual search mechanism to enhance MLLM's contextual understating capacities. In Table 3, to ensure a fair comparison with $V^*$ [8], we present our method's results using the experimental settings from [8]. We also report the accuracy metrics (1st and 2nd rows) for the multimodal benchmarks as adopted by LLAVA [9] in its paper, As stated in V* [8] and evident from the Table 3, there's significant degraded performance on MM-Vet [10], LLaVA-Bench$^W$ [11] and MMBench [12]. In contrast, our method demonstrates consistent improvements across all these multimodal benchmarks. ViperGPT [13] also utilizes LLM to solve visual tasks, however, it focuses on code generation to solve complex tasks based on Codex, which isn't related with our method.

## D VISUALIZATION RESULT ANALYSIS

We've provided more visualization results in Table 4, 5, 6, and 7. Compared to LLaVA-1.5 7B [9], our method generates more reasonable and accurate responses to the questions.

As shown in Table 4, our approach can generate accurate movie titles, such as "The Godfather", and the two actors' names, such as "Al Pacino" and 'Robert De Niro". Additionally, it provides a corresponding introduction, such as "The movie is a classic crime drama film directed by Francis Ford Coppola, known for its iconic characters, storytelling, and memorable scenes" for the left example. In the right example, our method generates the precise title "The Lord of the Rings: The Fellowship of

---

[1] https://huggingface.co/WhereIsAI/UAE-Large-V1

Table 1: The ablation study of using an object detector or a panoptic segmentation model to extract object regions for pixel-level textual embeddings.

| Method | Region Generator | VQAv2 | GQA | VisWiz | SQA$^I$ | VQA$^T$ | MME-P | MME-C | MMB | MM-Vet | POPE | MMMU |
|---|---|---|---|---|---|---|---|---|---|---|---|---|
| Mipha-3B | - | 81.3 | 63.9 | 45.7 | 70.9 | 56.6 | 1488.9 | 295.0 | 69.7 | 32.1 | 86.7 | 32.5 |
| Mipha-3B$^+$ | GroundingDINO | 82.0↑ | 64.9↑ | 46.4↑ | 71.4↑ | 57.2↑ | 1491.7↑ | 350.2↑ | 71.0↑ | 34.5↑ | 88.4↑ | 32.9↑ |
| **Mipha-3B$^+$ (Ours)** | **OpenSeed** | **82.4**↑ | **65.3**↑ | **47.0**↑ | **71.8**↑ | **57.8**↑ | **1501.2**↑ | **369.1**↑ | **71.5**↑ | **35.1**↑ | **88.7**↑ | **33.5**↑ |

Table 2: The ablation study of fine-tuning with and without the proposed visual prompt. The first (Mipha-3B), second (Mipha-3B$^*$) and third (Mipha-3B$^+$) rows mean Mipha-3B baseline, fine-tuning on Mipha-3B without and with the proposed visual prompt using LoRA [14].

| Method | Visual Prompt | VQAv2 | GQA | VisWiz | SQA$^I$ | VQA$^T$ | MME-P | MME-C | MMB | MM-Vet | POPE | MMMU |
|---|---|---|---|---|---|---|---|---|---|---|---|---|
| Mipha-3B | - | 81.3 | 63.9 | 45.7 | 70.9 | 56.6 | 1488.9 | 295.0 | 69.7 | 32.1 | 86.7 | 32.5 |
| Mipha-3B$^*$ | ✗ | 81.4↑ | 64.3↑ | 45.9↑ | 71.0↑ | 56.5↓ | 1489.2↑ | 303.2↑ | 70.4↑ | 33.5↑ | 87.4↑ | 32.6 |
| **Mipha-3B$^+$ (Ours)** | ✓ | **82.4**↑ | **65.3**↑ | **47.0**↑ | **71.8**↑ | **57.8**↑ | **1501.2**↑ | **369.1**↑ | **71.5**↑ | **35.1**↑ | **88.7**↑ | **33.5**↑ |

Table 3: The results compared with V$^*$ [8] has been reported, we adopt the setting and benchmarks from V$^*$ [8], the compared benchmarks including MME [7], MMBench [12], SEED-Bench(Img) [15], MM-Vet [10], LLaVA$^W$: LLaVA-Bench$^W$ [16] and POPE [17]. LLaVA-7B$^+$ denotes LLaVA (7B)$^+$ using presented REVIP method.

| Method | MME | MMBench | SEED-Bench(Img) | MM-Vet | LLaVA$^W$ | POPE |
|---|---|---|---|---|---|---|
| LLaVA (7B) [8] | 1051.2 | 34.4 | 41.8 | 30.4 | 62.6 | 76.5 |
| V$^*$ (7B) [8] | 1128.9↑ | 33.1↓ | 41.7↓ | 27.7↓ | 59.1↓ | 82.4↑ |
| **LLaVA (7B)$^+$ (Ours)** | **1156.4**↑ | **37.5**↑ | **43.2**↑ | **34.5**↑ | **64.5**↑ | **82.8**↑ |

the Ring" and provides an accurate introduction, such as "The movie is an epic fantasy adventure that follows the journey of a fellowship of characters". In contrast, LLaVA's responses are very general without fully understanding the global contexts within the images.

In Table 5, our method not only recognizes the text "Me: I'll do it at 8. Time: 8.05. Looks like I gotta wait till 9 now", but also understands its implication, such as "the puppet is aware of the time and is intentionally delaying any work or task it might be assigned", in the left image. For the right image, our method generates more potential options, such as a "private school", "a religious school", etc. In contrast, LLaVA's answers are either inappropriate or too limited.

Finally, as shown in Table 6 and 7, our method understands spatial relationships and accurately recognizes the text within the images. For example, in the left example of Table 6, our approach accurately names the person on the left as "Keira Knightley". It also recognizes the plate number "S3302CD" in the left example of Table 7, while LLaVA's answers are all incorrect. This indicates the strong fine-grained multimodal understanding capacity of our proposed method.

**Visual input example, context-related questions:**

| | | |
|---|---|---|
| | 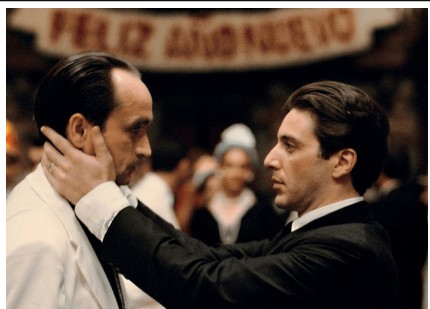 | 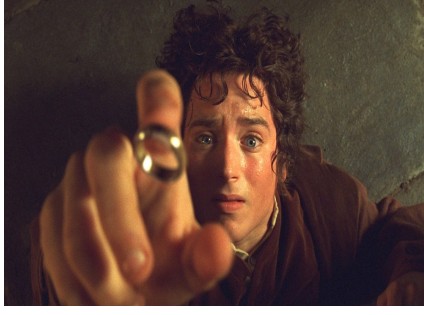 |
| | Can you give a short introduction to this movie? | Can you give a short introduction to this movie? |
| LLaVA-7B | The image features a scene from a movie where two men are standing close to each other, with one man placing his hand on the other's shoulder. They are both wearing tuxedos, which suggests that they are attending a formal event or a special occasion. The men appear to be engaged in a conversation or interacting with each other in some way, possibly discussing the event or sharing a moment of camaraderie. | The image features a young man wearing a brown shirt and holding a ring in his hand. He appears to be in a dark setting, possibly a cave or a dungeon. The man is looking at the camera, possibly engaging in a conversation or sharing a story. The ring he is holding suggests that the scene might be related to a fantasy or adventure movie, where the ring could represent a magical or powerful object. |
| **Ours** | The image features a scene from the movie "The Godfather", where two men played by actors Al Pacino and Robert De Niro, are standing close to each other. One man has his hand on the other's shoulder, and they appear to be engaged in a conversation or a confrontation. The movie is a classic crime drama film directed by Francis Ford Coppola, known for its iconic characters, storytelling, and memorable scenes. | The image features a scene from the movie "The Lord of the Rings: The Fellowship of the Ring", where the character Gandalf is holding out his hand to the hobbit Frodo, offering him the One Ring. This iconic moment captures the essence of the story, where Frodo is entrusted with the task of destroying the ring to prevent its power from falling into the wrong hands. The movie is an epic fantasy adventure that follows the journey of a fellowship of characters as they embark on a perilous quest to rid Middle-earth of the dark lord Sauron and his army of orcs. |

Table 4: The challenging examples on LLaVA-1.5, our approach can provide reasonable reponses for the tricky questions which need to fully understand the contexts of the images.

**Visual input example, context-related questions:**

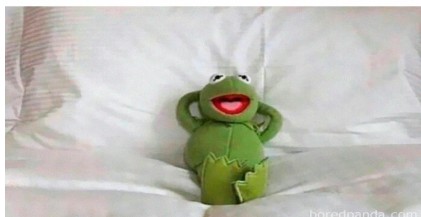

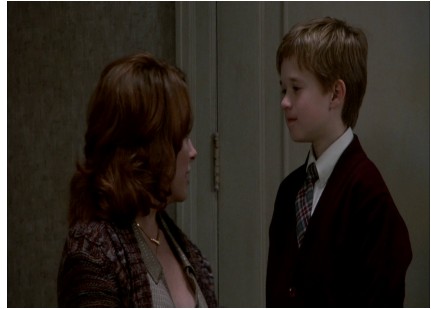

| | Can you explain this meme? | What kind of school does the boy go to? Answer it and give the rationale. |
|---|---|---|
| LLaVA-7B | The meme features a green Kermit the Frog puppet lying on a bed, with a caption that reads, "Me looks like I gotta wait till 9" The puppet appears to be relaxing on the bed, possibly waiting for someone or something. The image is a playful and lighthearted representation of a common situation where people might have to wait for a certain time before they can proceed with their plans. | The boy likely goes to a private school, as he is wearing a suit and tie, which is a common attire for students in private schools. The presence of a tie suggests that the school has a formal dress code, and the boy is dressed appropriately for the occasion. |
| **Ours** | The image features a green frog, which is a popular character from the television show "The Muppet Show", the frog is sitting on a bed, appearing to be in a relaxed and comfortable position. The puppet is holding its belly and seems to be enjoying its time. The bed has a white sheet, and the scene appears to be set in a cozy bedroom. The meme is captioned with a humorous message which reads "Me: I'll do it at 8. Time: 8.05. Looks like I gotta wait till 9 now", this implies that the puppet is aware of the time and is intentionally delaying any work or task it might be assigned. The meme is likely meant to be amusing and entertaining, as it combines the idea of a puppet with a human-like sense of humor and awareness of time. | The boy in the image is wearing a tie, which suggests that he might be attending a school with a formal dress code or a school that requires students to wear ties as part of their uniform. This could be a private school, a religious school, or a school with a strong emphasis on discipline and professionalism. The presence of a woman, who is wearing a dress, in the image, possibly his mother, indicates that the boy might be going to school with her support and guidance. |

Table 5: The challenging examples on LLaVA-1.5, our approach can provide reasonable reponses for the tricky questions which need to fully understand the contexts of the images.

**Visual input example, relationship-aware questions:**

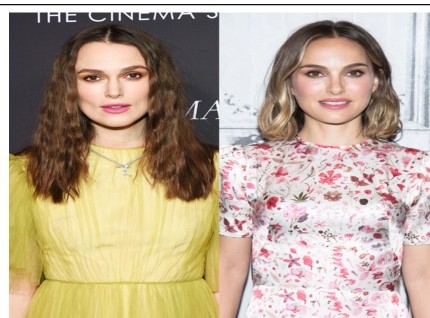 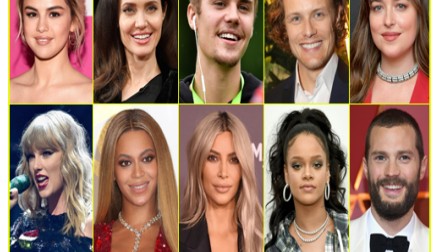

| | Who is the person on the left? | Who is the person in the bottom right? |
|---|---|---|
| LLaVA-7B | The person on the left is Reese Witherspoon. | The person in the bottom right is a man named Tom Hiddleston. |
| **Ours** | The person on the left is Keira Knightley. | The person in the bottom rights is Jamie Dornan. |

Table 6: The challenging examples on LLaVA-1.5, our approach can provide reasonable reponses for the tricky questions which need to fully understand the relationships among objects within images.

**Visual input example, text-related questions:**

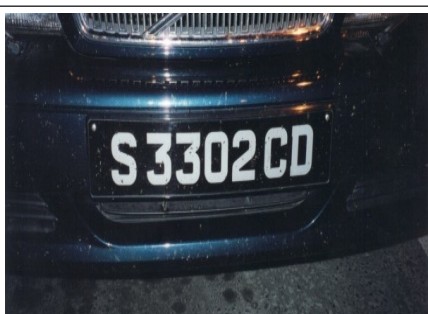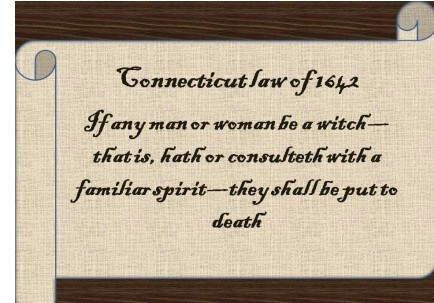

| | What are all the scene text in the image? | What are all the scene text in the image? |
|---|---|---|
| LLaVA-7B | The scene text in the image is "S33020CD". | The scene text in the image is "Connecticut Law of 1652". |
| **Ours** | The scene text in the image includes the license plate number "S3302CD" which is written in white on a black background. | The scene text in the image includes "Connecticut Law of 1642", which states "If any man or woman be a witch - that is hath or consulteth with a familiar spirit - they shall be put to death". |

Table 7: The challenging examples on LLaVA-1.5, our approach can provide reasonable reponses for the tricky questions which need to accurately recognize the texts within the images.