# OpenReview forum: "REVIP: Rethinking Visual Prompting for Multimodal Large Language Models with External Knowledge"
_ICLR.cc/2025/Conference — ICLR 2025 Conference Withdrawn Submission_

### Official Review · Reviewer_Ykur · 2024-10-31

**Soundness:** 2
**Presentation:** 2
**Contribution:** 2
**Rating:** 5
**Confidence:** 4

**Summary:**

This paper introduces a novel visual prompt approach that directly embeds fine-grained external knowledge into MLLMs, improving their ability to understand detailed visual elements. Specifically, the proposed method extracts pixel-wise contextual information using panoptic segmentation and OCR models and combines it with visual features, enabling the model to better understand fine-grained local objects and the overall image context.

**Strengths:**

1.	This paper points out that explicitly integrating region coordinate information into text prompts increases the complexity of model learning. The model needs to understand those coordinates and establish connections with visual content.

2.	This paper proposes integrating contextual embeddings within local contours as visual prompts to help establish correlations between image pixels and contexts.

3.	The paper is clearly articulated and Figures 1 and 2 are helpful to understand the methods.

**Weaknesses:**

1.	3.	When comparing to other visual prompt paradigms, only methods based on relevant region cropping are considered. It is recommended to include more comparisons with other visual instruction tuning methods, like the text-based coordinate representation scheme mentioned in the introduction, referring to [1]-[3].

[1] Visual instruction tuning[J]. Advances in neural information processing systems, 2024.

[2] Visionllm: Large language model is also an open-ended decoder for vision-centric tasks[J]. Advances in Neural Information Processing Systems, 2024.

[3] Minigpt-v2: large language model as a unified interface for vision-language multi-task learning, 2023.

2. The novelty seems limited. Moreover, using mask as visual prompt is not that reasonable.  When understanding  a complex scene, how to accurately mask irrelvant parts? Moreover, why not apply this method to typical image recgonition tasks, such as classification? Specifically,  it cloud be interesting to show how they handle complex scenes with potentially overlapping or ambiguous masks. Additionally, inquiring about the method's applicability to standard image recognition tasks would provide valuable insight into its broader potential.

**Questions:**

1.	When combining the image tokens and the processed auxiliary visual prompt, are the embeddings concatenated along dimension dv or dimension Nv? If they are concatenated along dv, why is it mentioned that "the total number of image tokens remains unchanged through the mapping"?

2.	Mathematically, feature addition is a special case of feature concatenation followed by an identity matrix production.  In other words,  feature concatenation is a nore general case of feature addition. However, according to the results reported in Table 1, the feature addition shows better performance compared to the feature fusion. Could you provide further explanation for this?

3.	The two images in Table 6 seem to be misplaced.

4.	The bolded numbers in Table 7 do not seem to indicate the optimal value. What is the specific meaning of the bolding for Qwen-VL on VQAT?

---

### Official Review · Reviewer_S9gM · 2024-11-01

**Soundness:** 3
**Presentation:** 3
**Contribution:** 2
**Rating:** 5
**Confidence:** 3

**Summary:**

This paper proposes to improve multimodal language models by using the contextual information in the panoptic segmentation masks and OCR bounding boxes. The experiment results demonstrate that the proposed method can improve MLLMs’ capabilities.

**Strengths:**

1. This paper proposes to use panoptic segmentation masks and OCR bounding boxes to inject external knowledge to the visual prompt, which is different from existing methods that integrate region coordinates information into text prompt. The proposed visual prompt provides more direct visual information.
2. The experiment results demonstrate the effectiveness of the proposed method to enhance the MLLMs’ capabilities.

**Weaknesses:**

1. The proposed idea to use panoptic segmentation masks and OCR bounding boxes to enhance the visual prompt is very straightforward. The contribution of this paper is not enough.
2. Although introducing external information like panoptic segmentation masks bring performance improvement, it needs a segmentation model to extract masks, which increases the computational complexity of the data preparation. On the other hand, in practical application when inference, it seems that users are required to provide the image and its panoptic segmentation, which is unreasonable.
3. The sensitivity of MLLMs’ capabilities to the extracted panoptic segmentation masks needs to be discussed. How does the quality of the masks influence the performance of the MLLMs?

**Questions:**

1. As mentioned above, the impact of the panoptic segmentation mask quality on the performance of MLLMs should be discussed.
2. Besides the panoptic segmentation masks, will other forms of external information, such as depth maps, further enhance the capabilities of MLLMs?

---

### Official Review · Reviewer_ZEjH · 2024-11-04

**Soundness:** 3
**Presentation:** 3
**Contribution:** 2
**Rating:** 5
**Confidence:** 4

**Summary:**

This paper explores integrating localized information into multimodal large language models (MLLMs) and introduces a new visual prompting method (named REVIP) that integrates contextual embedding within local contours (i.e., object masks) as the visual prompt. The visual prompt is then fused with the original visual features in a pixel-wise manner. The proposed REVIP method can be easily combined with existing MLLMs shows consistent performance improvement on Mipha-3B, LLaVA-1.5 7B and 13B across 11 benchmarks.

**Strengths:**

+ The overall paper is well written and easy to follow. The motivation and core idea are clear.
+ The proposed visual prompting method REVIP is simple yet effective, which shows consistent performance improvement on 11 different benchmarks. Moreover, the proposed method is applicable to various MLLMs.
+ Comprehensive ablation experiments and analysis validate the effectiveness of the proposed method.

**Weaknesses:**

- The authors claim that ‘MLLMs trained with image-text pairs still often struggle in fine-grained multimodal comprehension capacities, e.g., correctly counting objects or precisely locating a specific object’ and introduce panoptic segmentation masks to alleviate such issue. However, most of the benchmarks are VQA benchmarks. I am wondering, whether the proposed method can improve or unleash the counting and localization capabilities of MLLMs on REC/RES tasks?
- The utilized panoptic segmentation masks have inherent limitations. For instance, as the vocabulary of the segmentation models are pre-defined, it is hard to scale to more categories and domains. In addition, the granularity of the mask is also pre-defined. Thus, it is hard to handle more fine-grained cases such as human/object parts. I am also wondering how do the model decide whether to use segmentation or OCR method during inference?
- Some relevant visual prompting works are missing, which should be discussed and compared.

  [a] Set-of-Mark Prompting Unleashes Extraordinary Visual Grounding in GPT-4V.

- Some figures can be further improved. In Figure 3, it would be better to show which parts are trained or fixed. In Table 6, the two figures are in incorrect order.

**Questions:**

Please refer the Weaknesses.

---

> ### Comment · Reviewer_ZEjH · 2024-11-26
>
> As the authors have not provided any response to address my concerns, I would like to keep my negative rating.

---

### Official Review · Reviewer_Hpu6 · 2024-11-04

**Soundness:** 2
**Presentation:** 3
**Contribution:** 2
**Rating:** 5
**Confidence:** 4

**Summary:**

The paper proposed to extract the semantic information of the image, as one visual prompt to enhance the ability of MLLM.
One pipeline for generate the auxiliary visual prompt generation is proposed, where the panoptic segmentation and OCR model is employed.
Experimental results on different MLLMs demonstrate the effectiveness.

**Strengths:**

- The paper propose one novel perspective to rethink the image representation other than the existing work. One thread of work is converting the local context information into text prompts. The proposed method rely on panoptic segmentation and OCR detection to convert image into visual tokens.
- The proposed method can be easily accomodate with the existing MLLM, which thereby improve the corresponding performances.

**Weaknesses:**

- The proposed method seems to be very similar to existing work [1], which rely on different visual encoders to enhance the visual representation of the image. The difference is that the proposed method carefully design one pipeline relying on panoptic segmentation, OCR detection to yield the other visual tokens of the image. Please discuss the underlying technical difference of the proposed methods with the aforementioned methods.
[1] DeepSeek-VL: Towards Real-World Vision-Language Understanding

- It seems that the stated visual prompt highly relies on the performances of panoptic segmentation and OCR detection. I am wondering the effects of the accuracy of the proposed auxiliary visual prompt generation. If we use different panoptic segmentation methods and different OCR detection methods, what about the performances of the auxiliary visual prompt as well as the MLLM performances.

- The proposed method only performs in the SFT stage. What about the performance gain if the proposed method is also performed in the pre-training stage. Another related question is that if performed in the pre-training stage, all the images should undergo the auxiliary visual prompt generation. The proposed pipeline seems to be suitable for the natural iamges. what about the other types of images, such as the remote sensing images, the compond image (document image).

**Questions:**

Please check the detailed information in the Weaknesses part.

---

> ### Comment · Reviewer_Hpu6 · 2024-11-26
> **Review comments**
>
> The authors do not provide any rebuttal information. Therefore, I will keep my previous rating.

---

### Note · Authors · 2024-11-26

I have read and agree with the venue's withdrawal policy on behalf of myself and my co-authors.